# CMT-BENCHMARK: A BENCHMARK FOR CONDENSED MATTER THEORY BUILT BY EXPERT RESEARCHERS

**Haining Pan**
Department of Physics and Astronomy
Rutgers University

**James V. Roggeveen**
School of Engineering and Applied Sciences
Harvard University

**Erez Berg**
Department of Condensed Matter Physics
Weizmann Institute of Science

**Juan Carrasquilla**
Department of Physics
ETH Zürich

**Debanjan Chowdhury**
Department of Physics
Cornell University

**Surya Ganguli**
Department of Applied Physics
Stanford University

**Federico Ghimenti**
Department of Applied Physics
Stanford University

**Juraj Hasik**
Department of Physics
University of Zürich

**Henry Hunt**
Department of Applied Physics
Stanford University

**Hong-Chen Jiang**
Stanford Institute for Materials and Energy Sciences
SLAC National Accelerator Laboratory

**Mason Kamb**
Department of Applied Physics
Stanford University

**Ying-Jer Kao**
Department of Physics
National Taiwan University

**Ehsan Khatami**
Department of Physics and Astronomy
San José State University

**Michael J. Lawler**
Department of Physics
Cornell University

**Di Luo**
Department of Electrical and Computer Engineering
University of California, Los Angeles

**Titus Neupert**
Department of Physics
University of Zürich

**Xiaoliang Qi**
Department of Physics
Stanford University

**Michael P. Brenner**
School of Engineering and Applied Sciences
Harvard University
Google Research

**Eun-Ah Kim**
Department of Physics
Cornell University
ek436@cornell.edu

## ABSTRACT

Large language models (LLMs) have demonstrated remarkable progress in coding and mathematical problem-solving; however, evaluation on advanced research-level problems in the hard sciences remains scarce. To fill this gap, we present **CMT-Benchmark**, a dataset of 50 original problems covering condensed matter theory (CMT) at the level of an expert researcher. The solution for these problems involve analytical and computational approaches commonly used in quantum many-body physics and classical statistical mechanics. The dataset has been designed and verified by a worldwide panel of expert researchers through a

collaborative environment. Topics in the dataset include Hartree-Fock mean-field theory, exact diagonalization methods, quantum Monte Carlo sampling, density matrix renormalization group, quantum statistical mechanics, classical statistical mechanics, and model building. We evaluate different LLMs by programmatically checking LLM-generated solutions against expert-supplied ground truth. To verify LLMs performance at scale, we developed an automated machine-grading pipeline suitable for advanced physics research problems. For example, we handle non-commuting operators that are essential for quantum many-body problems by symbolic manipulation and normal ordering. Our evaluations show that frontier models struggle with all of the problems in the dataset, highlighting a gap in the physical reasoning skills of current LLMs. Notably, experts identified strategies for creating increasingly difficult problems by interacting with the LLMs and exploiting common failure modes. While the highest-performing model, GPT5, correctly solves 30% of the problems, average performance across 17 models (GPT, Gemini, Claude, DeepSeek, and Llama classes) is only $11.4\pm2.1\%$. Moreover, our benchmark contains 18 problems that *not a single one* of the 17 models considered here can correctly solve, and 26 problems that are solved by *at most* one model. These currently unsolvable problems span the fields of Quantum Monte Carlo, Variational Monte Carlo, and Density Matrix Renormalization Group. Furthermore, we illustrate how incorrect answers sometimes violate fundamental symmetries or have unphysical scaling dimensions. We believe that this benchmark set provides valuable guidance for the future development of language models, aiming to achieve the goal of AI research assistants and tutors.

# 1 INTRODUCTION

The progress of Frontier LLMs has been stunning. Whereas a few years ago models struggled on high school mathematics problems (Hendrycks et al., 2021), today's LLM-based systems achieve Gold medals in the International Math Olympiad (Trinh et al., 2024) and competitive coding competitions, inventing solutions that humans are unable to discover. Benchmarks for assessing LLMs against expert-level mathematics have flourished (Liu et al., 2024; Fan et al., 2024; Roggeveen et al., 2025), including expert-level benchmarks made by professional mathematicians (Glazer et al., 2025). At the same time, there is an intense interest in LLMs for science, with a significant literature focused on creating benchmarks to evaluate agent capabilities against hard science problems (Laurent et al., 2024; Mitchener et al., 2025; Feng et al., 2025; Cui et al., 2025; Pan et al., 2025; Pramanick et al., 2025). However, existing hard science benchmarks measure knowledge or skill for carrying out textbook problems for students at varying levels, and do not assess whether models can function as a research assistant on cutting-edge scientific tasks. Typical crowdsourcing strategies will not work in highly technical fields like theoretical condensed matter physics, since the required expertise is focused on small communities. While the mathematics community built such benchmarks by assembling groups of leading experts to create research-grade problems (Glazer et al., 2025; Balunović et al., 2025), a systematic counterpart is missing in hard scientific domains.

We created **CMT-Benchmark** to address this gap so that the LLM research community can hill-climb towards a competent AI research assistant. For LLMs to serve as scientific research assistants, they must demonstrate rigorous critical judgment and the ability to synthesize existing knowledge with theoretical principles established in a specific scientific domain. We designed the problems and a rigorous yet automated evaluation scheme, focusing on condensed matter theory (CMT), a subfield of physics reveals how collective interactions among particles generate emergent phenomena such as superconductivity and topological phases, while also providing the theoretical foundation for advanced materials and quantum technologies. Research of CMT requires synthesizing microscopic knowledge of material systems with macroscopic observations in a manner that adheres to the theoretical principles. **CMT-Benchmark** consists of 50 original, select, high-value problems covering seven computational and theoretical methods, as well as sound model-building (categorized as "Other"), as shown in Fig. 1a. The novelty of our problems lies in the design principles we adopted from the principles of trustworthy and impactful scientific research. For this, we assembled an international panel of expert researchers to write original problems and provide critiques of each other's problems. Each contributor crafted problems they would expect a strong graduate student or research assistant to

answer correctly, measuring critical skills for performing research in their field. They then iteratively refined the problems to identify gaps in critical judgments and insights in LLM reasoning.

The design principle of our evaluation scheme is also based on the mission of aiding the development of a competent AI research assistant. Scientific research must push the knowledge frontier so that other researchers in the community can build on the outcome. Correctness must be absolute, and results should be deterministically reproducible. Hence, unlike the typical homework grading setting where the grader issues partial credit, we apply the rigorous standards we hold ourselves to: we demand that the answers be deterministically and objectively correct. We designed problems in multiple answer formats that can be automatically graded, including multiple-choice, numerical values, algebraic expressions, and non-commutative operator expressions. We score the LLM-generated solutions as correct or incorrect against the ground truth answer supplied by the author of the problem. Even the most advanced models exhibit low performance on our problems, with the highest pass rate of 30%. Our results reveal that current LLMs cannot function as research assistants.

We highlight that **CMT-Benchmark** makes the following important contributions:

1. **Benchmark for Analytic and Computational Reasoning.** It is the first benchmark explicitly designed to jointly test analytic and computational reasoning in LLMs, assessing their potential as scientific research assistants in CMT—a field central to understanding emergent quantum phenomena and foundational to quantum materials and technologies.
2. **High-Value, Expert-Curated Research Level Dataset.** Our dataset comprises 50 original and rigorously designed problems spanning seven computational and theoretical methods, plus model-building. Problems were authored and refined by an international panel of expert researchers, including postdocs and professors in top universities, ensuring that each reflects the level of reasoning expected from a strong graduate student or research assistant.
3. **Rigorous Evaluation Revealing Fundamental Gaps in LLM Reasoning.** Even frontier models struggle on **CMT-Benchmark**: GPT-5 solves only 30%, with the average across 17 models at 11.4±2.1%. Moreover, 18 problems are unsolved by any model, and 26 are solved by at most one. We further diagnose common failure modes, including violations of fundamental symmetries and unphysical scaling dimensions, highlighting critical reasoning gaps and establishing **CMT-Benchmark** as a roadmap for advancing AI scientific assistants.

## 2 RELATED WORKS

Progress in evaluating expert-level scientific reasoning has been lagging behind mathematical reasoning. The standard metric for measuring scientific prowess remains the 2023 benchmark Graduate-Level Google Proof Question and Answer (GPQA) (Rein et al., 2023), although the performance of Frontier LLMs has nearly saturated. Recently, Humanity's Last Exam (HLE) (Phan et al., 2025) raised the bar, with a crowdsourced collection that incentivized hard problems, with some fraction focusing on the sciences, spread across a wide range of categories. Although the difficulty of the benchmark is appealing, neither HLE nor any of the existing benchmarks measure the qualification of LLM to serve as research assistants in **specific** scientific domains.

Nevertheless, a recent benchmark, (Wang et al., 2025), is notable as it focused on condensed matter physics textbook problems, with difficulty levels ranging from undergraduate to advanced graduate coursework. The dataset contains calculation problems extracted from textbooks on topics spanning Magnetism, Superconductivity, Strongly Correlated Systems, and Semiconductors. The LLM responses were evaluated using abstract syntax trees to offer partial credit, using a custom metric dubbed SEED, which exhibits 90% correlation with human experts. While impressive, this benchmark focuses on questions for students, rather than those at the cutting edge of research.

### 2.1 THE NEED FOR CONDENSED MATTER THEORY

We choose CMT both because of the importance of the domain and of its underrepresentation in widely used benchmarks such as HLE. As the largest branch of modern physics, condensed matter provides the foundation for understanding emergent quantum phenomena in materials and underpins transformative advances in quantum materials, quantum computation, and quantum technologies. The domain's close ties to material science, chemistry, and quantum technologies imply a high demand for AI research assistants in this field. However, the subject is challenging to teach algorithmically. The

Table 1: Comparison of selected advanced benchmarks. We list focus, sourcing method, evaluation, and size.

| Dataset | Focus | Problem Sourcing | Evaluation Method | Size |
|---|---|---|---|---|
| GPQA (Rein et al., 2023) | Graduate-level science Q&A (biology, physics, chemistry) | Domain experts | Multiple-choice accuracy (google-proof) | 448 |
| Humanity's Last Exam (Phan et al., 2025) | Broad coverage (multiple-choice + short answer) | Subject-matter experts worldwide | Automated grading for multiple-choice and short-answer | 2,500 |
| MathArena (Balunović et al., 2025) | Olympiad-style math competitions (exact answers) | Expert creation | Automated formula parsing | 96 |
| SciCode-Bench (Tian et al., 2024) | Scientific code generation across natural sciences | Scientist-curated research scripts | Unit tests and domain-specific test cases | 80 |
| TPBench (Chung et al., 2025) | Theoretical physics (high-energy, cosmology) | Researcher-authored novel problems | Combination of auto-verifiable checks and rubric-driven grading | 57 |
| PhySense (Xu et al., 2025) | Principle-based physics reasoning (theoretical physics) | Expert curation | Automated grading for multiple-choice and evaluation of token efficiency | 380+ |
| CMPhysBench (Wang et al., 2025) | Condensed matter physics (calculation problems) | Expert curation | Expression-tree edit distance with partial credit (Scalable Expression Edit Distance) | 520+ |
| **CMT-Benchmark** | Condensed matter theory (numerical and analytical) | Expert panel authorship and curation | Automatic parsing with numerical and symbolic equivalence | 50 |

theoretical framing of CMT involves modeling many interacting entities in accordance with a strict set of physical rules. Problems that quickly reach the limits of computational complexity abound. Cutting-edge research in the area requires a multifaceted approach that combines mathematics, theoretical formalisms such as field theory and non-commutative operator algebra, computational methods, a geometric understanding of the physical system, and fundamental concepts, including notions of symmetry. LLMs' tremendous progress in coding and mathematics, along with their possession of knowledge far exceeding that of an individual human, suggests that an AI research assistant may be attainable.

## 2.2 INNOVATIONS IN BENCHMARK CREATION

**CMT-Benchmark**, with its aim to evaluate research-readiness, goes beyond textbook knowledge or skills and applies a rigorous standard of correctness. In research, it is crucial to discern what not to do, because whether a meaningful answer can be reached is not known a priori. Our problems test such judgments and the ability to synthesize knowledge and skills to define and solve a meaningful problem. Moreover, a research output must be absolutely correct and reproducible. Hence, our problems are designed to be deterministically evaluated with a binary outcome. Since it took experts hours to write original problems, the number of problems in **CMT-Benchmark** is small compared to crowd- or textbook-sourced benchmarks. However, it is the first hard science equivalent of FrontierMath.

Another innovation we introduced is automatic parsing that can handle equivalent expressions of non-commuting operators. Our evaluation scheme was built on the framework for automatically parsing LaTeX introduced in (Roggeveen et al., 2025). However, with problems in quantum many-body physics, we needed the parser to recognize equivalent expressions based on operator algebra correctly. The formalism of quantum mechanics uses the algebra of non-commutative operators. A competent researcher can readily dismiss an operator expression as violating fundamental principles, one of the basic approaches for checking the "sanity" of calculations. While LLMs can carry out algorithmic manipulations, it is a different skill to inspect problems, statements, and reasoning from the fundamental principles underlying the formalism. Problems requiring operator expressions in their answer test such skills. Our parser handles operator algebra and identifies equivalent expressions, through symbolic manipulations and normal ordering, as described in Section 3.1.1.

## 3 DATASET

This dataset was constructed by an international panel of condensed matter theorists including postdocs and faculties from top universities, who were tasked to contribute original problems they would expect their group members to answer correctly. We required the problems to be unambiguous and lead to a single, verifiable solution that could be parsed and machine-graded by our evaluation software. In addition to the problem and solution, authors were required to provide a written explanation of their solution to facilitate easy verification of the problem's correctness by other panel members. Our dataset covers a broad range of solution modalities, including algebraic expressions, numerical values, multiple-choice questions, and operator expressions, as shown in Fig. 2. For

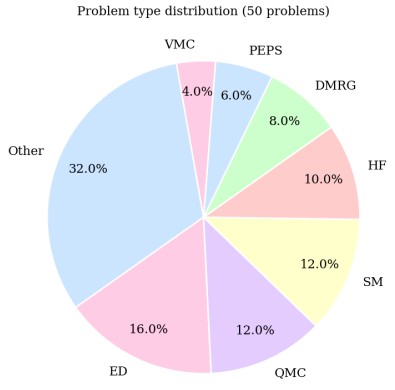

Problem type distribution (50 problems)

(a) Distribution of problem types in **CMT-Benchmark**.

| Type | Example question |
|------|------------------|
| ED | Which among $N$, $S^z$, $\eta^2$, $\sum_i n_{i\uparrow} n_{i\downarrow}$, and $\sum_{i,\sigma} c^\dagger_{i\sigma} c_{i+1,\sigma}$ are good quantum numbers for $H = -t \sum_{i,\sigma} (c^\dagger_{i\sigma} c_{i+1,\sigma} + \text{H.c.}) + U \sum_i n_{i\uparrow} n_{i\downarrow} + \sum_i h_i S^z_i$? |
| QMC | Which methods are provably efficient for 1D vs 2D ground-state properties given sign-problem constraints? |
| SM | What is the critical coupling $K_c$ for synchronization stability in coupled large-$N$ soft-spin systems? |
| HF | Which HF order parameters preserve translational symmetry on a 2D triangular lattice? |
| DMRG | What is the ground-state degeneracy of an open Kitaev alternating chain? |
| VMC | Which projections restore $C_4$ rotation symmetry of a $J_1{-}J_2$ variational wavefunction? |
| PEPS | In a momentum-superposed single-defect iPEPS excitation $H_k B = \omega_k f$, express $f$ using only $N_k$ and $B$. |

(b) Representative example questions by problem type.

Figure 1: Problem type distribution and representative example questions in each type.

multiple-choice problems, we ask for one or more choices from among many options (over 5 for most problems) to avoid 'lucky guesses'. All solutions were evaluated using a uniform evaluation framework with a strict passing standard of correctness without partial credit. Our dataset can be found in the Hugging Face repository (Huggingface, 2025).

## 3.1 PROBLEM SUBMISSION AND VERIFICATION PIPELINE

We built **CMT-Benchmark** using a Google Sheet running a custom-built extension. Using the extension the panel could test their solutions against the parsing infrastructure before evaluations, ensuring that ground truth solutions were compatible with our grading framework. The extension also allowed writers to run their prompts through a subset of the LLMs used in the final evaluation, including providing machine grading of the LLM solutions. The authors used this feature to increase the difficulty of their problems iteratively. This iterative approach also helped authors remove any ambiguities in their problems and significantly improve the quality of the dataset.

When a problem fails every model available on the sheet, another author would review the problem and solution for correctness before the problem was accepted into the final benchmark dataset. On the spreadsheet, authors had access to Gemini 2.0 Flash, Gemini 2.5 Flash, Gemini 2.5 Pro, and GPT-4o. This iterative problem-building approach, using a custom Google Sheet integration, mirrors the success of other recent benchmarks (Roggeveen et al., 2025).

### 3.1.1 INFRASTRUCTURE FOR AUTOMATING PARSING AND EVALUATION

To enable automated grading of mathematical expressions for correctness, we implement a LaTeX to Sympy parser that converts a raw expression provided by either an author or LLM into an expression that can be evaluated. The parser used in this benchmark builds on that used for standard algebraic problems previously used in mathematics benchmarks (Roggeveen et al., 2025). To enable parsing, authors are required to follow certain guidelines in formatting their answers, along with providing a list of the parameters, variables, and functions they expect to appear in their solution. These must all be defined in the prompt. The model is instructed to return its final answer in a boxed LaTeX environment and not to introduce any new variables as part of its solution. Although models may generate arbitrary intermediate reasoning, our evaluation pipeline deliberately discards these traces and evaluates only the final boxed expression, keeping the benchmark focused on final research conclusions while remaining compatible with future extensions that explicitly evaluate reasoning traces.

A novel component of the parsing logic for this benchmark was the introduction of non-commuting operators. While standard algebraic expressions may be evaluated by substituting scalar values for

---

**Answer modality: Numerical value**

**Question:** Consider a classical O(3) spin Hamiltonian in two spatial dimensions on a triangular lattice: $H = -\sum_{i,j} J_{ij} S_i \cdot S_j$, where $J_{ij} = J$ for x-directed bonds and $J_{ij} = J'$ otherwise. At $T = 0$, find the number of gapless Goldstone modes, $n_{FM}$, for ferromagnetic couplings ($J > 0, J > J' > 0$), and $n_{AF}$, for antiferromagnetic couplings ($J < 0, J < J' < 0$). Return $n_{FM}$ and $n_{AF}$ in a ☐ latex environment separated by a ;.

**Answer:** $\boxed{1\,;3}$

---

**Answer modality: Multiple choice**

**Question:** Consider the dynamics in two dimensions of the following modified active Brownian particle: $\dot{\mathbf{x}} = v_0 \mathbf{u}$, where $v_0$ is a positive constant, and $\mathbf{u}$ is a vector of unit norm, whose orientation $\theta$ with respect to the $x$ axis evolves according to the overdamped dynamics: $\dot{\theta} = -\frac{v_0}{T} \nabla V \cdot \mathbf{Au} + \chi(t)$, where $V(\mathbf{x})$ is an external potential that depends only on $\mathbf{x}$, and $T$ is a positive constant. The matrix $\mathbf{A} = \begin{bmatrix} 0 & -1 \\ 1 & 0 \end{bmatrix}$ is a fully antisymmetric two-dimensional matrix. Consider a perturbation of the potential $V \to V + h(t)\phi(\mathbf{x})$. Consider the steady state linear response function $R(s) = \langle \frac{\delta\phi(\mathbf{x}(t+s))}{\delta h(t)} \rangle$ and the steady state autocorrelation function $C(s) = \langle \phi(\mathbf{x}(t))\phi(\mathbf{x}(t + s)) \rangle$. Is the fluctuation dissipation theorem between these correlation and response functions violated? Choose one of the following options: (a) Yes, because the dynamics has a positive entropy production rate. (b) No, because the dynamics is time-reversible. (c) No, because the Boltzmann distribution is the stationary distribution. (d) Yes, because there is a nonzero self-propulsion speed. Return your choice among the options "a","b","c" and "d" enclosed in a ☐ LaTex environment.

**Answer:** $\boxed{c}$

---

**Answer modality: algebraic expressions**

**Question:** Consider a peculiar example of Kitaev alternating chain, whose Hamiltonian is given by $H = -\sum_i^{N/2}(\sigma_{2i-1}^x \sigma_{2i}^x + \sigma_{2i}^y \sigma_{2i+1}^y)$, where $\sigma_i^x$ and $\sigma_i^y$ are Pauli matrices on site $i$, and $N$ is the number of sites. Calculate its ground state degeneracy for an open chain in terms of $N$ and the value of the central charge $c$. Denote these degeneracies as a function of $N$ and the value of the central charge $c$, and return your answer in LaTeX as ☐.

**Answer:** $\boxed{2^{N/2-1}\,; c = 1/2}$

---

**Answer modality: non-commutative operator expressions**

**Question:** Consider the Fermi-Hubbard Hamiltonian with nearest-neighbor hopping $t$ in its particle-hole symmetric form on a bipartite lattice with a chemical potential term. Express the Hamiltonian after the following transformation: $c_{i,\uparrow}^\dagger = p_{i,\uparrow}^\dagger$ and $c_{i,\downarrow}^\dagger = \pm p_{i,\downarrow}$, depending on whether $i$ is on the A sublattice or the B sublattice, taking $m_{i,\sigma}$ to be the new density operator. The answer will take the form of $H = \sum_{\langle i,j \rangle, \sigma} f_{i,j,\sigma} + \sum_i g_i$, where the only operators in $f_{i,j,\sigma}$ are the $p$ operators and the only operators in $g_i$ are the $m$ operators. Return the expression for $f_{i,j,\sigma} + g_i$ in a ☐ LaTeX environment. Your answer should not include any $\sum$ notation or the Hermitian conjugate (H.c.) abbreviation.

**Answer:** $\boxed{-t(p_{i,\sigma}^\dagger p_{j,\sigma} + p_{j,\sigma}^\dagger p_{i,\sigma}) - U(m_{i,\uparrow} - \frac{1}{2})(m_{i,\downarrow} - \frac{1}{2}) - \mu(m_{i,\uparrow} - m_{i,\downarrow}) - \mu}$

Figure 2: Example questions in **CMT-Benchmark** by four answer modalities: numerical value, multiple choice, algebraic expressions, and non-commutative operator expressions.

variables and evaluating a single numeric value, non-commuting operators are not amenable to this treatment. As these non-commuting operators play a key role in quantum condensed matter physics problems (e.g., Hamiltonians $H = \sum_{ij} t_{ij} c_i^\dagger c_j + U \sum_i n_i$), a benchmark incapable of correctly evaluating these expressions would be missing a core component of the field.

We handle such expressions by having authors declare whether any such expressions exist in the problem. These declarations are known only to the parser and are not passed to the model for its evaluation. In these cases, we replace any non-commutative expression with a non-commutative Sympy symbol and then invoke standard physics simplifications (e.g., $\{c_i, c_j^\dagger\} = \delta_{ij}$ for fermions) to reduce both the specified solution and the model's response to a canonical, order-sensitive form, such as the "normal ordering" from condensed matter. We then verify equivalence using standard Sympy symbolic equivalence checks. We will release the code to perform machine grading in the future.

## 3.2 PROBLEM TYPES

The dataset includes 8 problem types in terms of the contents, as shown in Fig. 1a, covering 7 computational and theoretical methods and sound model building. The computational and theoretical methods covered are Hartree–Fock (HF), Exact Diagonalization (ED), Density Matrix Renormalization Group (DMRG), Quantum Monte Carlo (QMC), Variational Monte Carlo (VMC), Projected Entangled Pair States (PEPS), and Statistical Mechanics (SM). Problems that test sound model-building and the use of fundamental principles are labeled as the *Other* type. Example questions from each problem type are shown in Fig. 1b and detailed descriptions follow. We employ a diverse range of answer formats, including algebraic expressions, numerical values, multiple-choice questions, and operator expressions, as illustrated in Fig. 2, to evaluate LLM's capabilities from multiple angles while ensuring automatic and deterministic evaluations.

### 3.2.1 HARTREE–FOCK (HF)

Problems cover self-consistent mean-field decouplings and ground-state characterization on lattices, classification of order parameters consistent with symmetries, Brillouin-zone folding under commensurate charge-density waves, and numerical complexity estimates for plane-wave representations. For example, we consider solving the self-consistency equation for Hartree–Fock mean-field theory on a 2D triangular lattice associated with the following mean-field Hamiltonian with mean-field terms being $H_{\text{Hartree}} = \frac{1}{N} \sum_{s,s'} \sum_{k_1,k_2} U(0) \langle c_s^\dagger(k_1) c_s(k_1) \rangle c_{s'}^\dagger(k_2) c_{s'}(k_2)$ and $H_{\text{Fock}} = -\frac{1}{N} \sum_{s,s'} \sum_{k_1,k_2} U(k_1 - k_2) \langle c_s^\dagger(k_1) c_{s'}(k_1) \rangle c_{s'}^\dagger(k_2) c_s(k_2)$, where $U(k) = \sum_n U_n e^{-ik \cdot n}$ is the repulsive interaction strength ($U_n > 0$) in the momentum basis. What are the possible order parameters that preserve translational symmetry for a Hartree–Fock mean-field Hamiltonian on a two-dimensional triangular lattice?

### 3.2.2 EXACT DIAGONALIZATION (ED)

Problems cover finite-size many-body spectra and symmetry resolution, including identification of good quantum numbers and block-diagonal sectors, counting symmetry-distinct momentum and point-group blocks, diagnosing exact versus asymptotic degeneracies, scaling of low-lying level spacings, small-cluster combinatorics for model building, and translational or gauge-structure consequences for expectation values and band minima. For example, consider a Hamiltonian for $N$ fermions, $H = -t \sum_{i,\sigma} \left( c_{i\sigma}^\dagger c_{i+1,\sigma} + \text{H.c.} \right) + \sum_i U\, n_{i\uparrow} n_{i\downarrow} + \sum_i h_i S_i^z$, and ask which of the following are good quantum numbers: (a) $N$; (b) $S^z$; (c) $\sum_{i,\sigma} c_{i\sigma}^\dagger c_{i+1,\sigma}$; (d) $\eta^2 = \frac{1}{2} \left( \eta^+ \eta^- + \eta^- \eta^+ \right) + (\eta^z)^2$, with $\eta_- = \sum_i (-1)^i c_{i\uparrow} c_{i\downarrow}$, $\eta_+ = \eta_-^\dagger$, and $\eta_0 = \frac{1}{2}(\hat{N} - L)$; (e) $\sum_i n_{i\uparrow} n_{i\downarrow}$.

### 3.2.3 DENSITY MATRIX RENORMALIZATION GROUP (DMRG)

Problems cover bond-dimension scaling versus system size, extraction and comparison of correlation lengths (e.g., bulk correlators versus boundary-pinned responses), effects of boundary conditions and geometry (chains, ladders, cylinders), and phase identification in concrete lattice models. For example, we consider a Kitaev alternating chain with Hamiltonian $H = -\sum_{i=1}^{N/2} \left( \sigma_{2i-1}^x \sigma_{2i}^x + \sigma_{2i}^y \sigma_{2i+1}^y \right)$, where $\sigma_i^x$ and $\sigma_i^y$ are Pauli matrices on site $i$ and $N$ is the number of sites, and ask for the ground-state degeneracy for an open chain in terms of $N$ and the central charge $c$.

### 3.2.4 QUANTUM MONTE CARLO (QMC)

Problems cover phase transitions in frustrated transverse-field Ising models on the triangular and 4–8 lattices (emergent $U(1)$ versus Ising behavior), sign-problem diagnostics in stochastic series expansion on square versus kagome lattices with $J_z$ and $J_{\pm\pm}$ terms, and determinant-QMC sign-problem conditions in fermionic settings (spinless fermions at half filling, a two-band model with onsite interactions and spin-mixing hoppings, and a long-range-interaction lattice model under specified densities, fields, and complex next-nearest-neighbor hoppings). For example, we consider the transverse-field Ising model in one dimension and on the 4–8 lattice with antiferromagnetic and strong ferromagnetic bonds arranged so that every plaquette carries $\pi$ flux. We ask which standard methods are provably efficient for computing ground-state properties in 1D and 2D for large system sizes (more than 200

spins), given the two-dimensional sign-problem constraints and the availability of Jordan–Wigner mappings, DMRG, transfer-matrix approaches, or simple variational constructions in one dimension.

### 3.2.5 VARIATIONAL MONTE CARLO (VMC)

Problems cover symmetry restoration/projection in lattice spin models and the correctness and variance of Monte Carlo estimators for neural-network wavefunctions. For example, we consider the 2D Heisenberg $J_1$–$J_2$ model with a wavefunction $\psi(x)$ that breaks rotation symmetry and ask which constructions restore $C_4$ rotation symmetry using the rotation operator $R$: (a) $\sum_n \psi(R^n x)$; (b) $\prod_n \psi(R^n x)$; (c) $\sum_n \psi(Rx)$; (d) $\sum_n (-1)^n \psi(R^n x)$; (e) $\sum_n e^{i\pi n^2} \psi(R^n x)$.

### 3.2.6 PROJECTED ENTANGLED PAIR STATES (PEPS)

Problems cover iPEPS excitation ansätze built by locally replacing a ground-state tensor and forming momentum superpositions, coarse-graining pipelines for extracting CFT data in classical 2D tensor networks, and SU(2)-symmetric iPEPS design with rotational ($C_4$) constraints and parameter counting. For example, we start from an iPEPS with ground-state tensor $A$, form a defect state by replacing one $A$ with $B$ at position $x = (i, j)$ so that $|\Psi_0(A)\rangle \to |\Phi(A, B)_x\rangle$, define the momentum superposition $|\Phi(B)_k\rangle = \sum_x e^{ik \cdot x} |\Phi(B)_x\rangle$, and consider the generalized eigenvalue problem $H_k B = \omega_k f$. We ask for the form of $f$ in terms of the normalization $N_k = \langle \Phi(B)_k | \Phi(B)_k \rangle$, using only $N_k$ and $B$.

### 3.2.7 STATISTICAL MECHANICS (SM)

Problems cover nonequilibrium stochastic dynamics and combinatorial models, including odd diffusivity in chiral active Ornstein–Uhlenbeck processes with inertia, fluctuation–dissipation checks for torque-driven active Brownian motion, synchronization thresholds in coupled random-tensor soft-spin networks, cavity-variable choices in the $d \to \infty$ limit for molecular liquids, Onsager–Machlup actions for multiplicative-noise Langevin equations, and counting fully packed dimers on cylinders with defect-density optimization. For example, consider two coupled systems with $N$ soft spins $\{x_i\}$ and $\{y_i\}$ obeying (Fournier et al., 2025) $\dot{x}_i = -\lambda(\mathbf{x})x_i + N^{-1}\sum_{j,k} J_i^{jk} x_j x_k + K(y_i - x_i)$ and $\dot{y}_i = -\lambda(\mathbf{y})y_i + N^{-1}\sum_{j,k} J_i^{jk} y_j y_k + K(x_i - y_i)$, where $\lambda(\mathbf{x}) = N^{-1}|\mathbf{x}|^2 - \gamma$ with $\gamma > 0$, and $J_i^{jk}$ is a symmetric random tensor of zero mean and variance $\sigma^2$. In the $N \to \infty$ limit and defining synchronization by $N^{-1}\sum_i (x_i - y_i)^2 = 0$ in steady state, we ask for the critical coupling $K_c(\gamma, \sigma^2)$ above which the synchronous state is stable.

### 3.2.8 OTHER

Problems cover model-building and application of fundamental principles: particle–hole mappings and operator rewrites in Hubbard-type models; strong-coupling correlators in dimerized chains; linked-cluster expansions via inclusion–exclusion; correlation decay and transition claims in long-range Ising models; transport and compressibility statements in frustrated boson models; and zero-temperature phase and correlation properties in quantum Ising–type systems. For example, we ask the LLM to choose from the following options for a classical Ising model with Hamiltonian $H = -\frac{1}{2}\sum_{i \neq j} J(|i - j|)\sigma_i\sigma_j$, where $J(n) = |J_0|/(1 + n^2)^\alpha$. We ask the LLM to choose from: (a) For $\alpha = 2$, there is a non-zero critical temperature $T_c$. (b) For $\alpha = 1$, at sufficiently low temperature, $\langle \sigma_j \sigma_{j+n} \rangle \to m^2 > 0$ as $n \to \infty$. (c) For $\alpha = 1$, there is a temperature for which $\langle \sigma_j \sigma_{j+n} \rangle - \langle \sigma_j \rangle\langle \sigma_{j+n} \rangle$ decays as the inverse of the logarithm of distance. (d) For $\alpha = 4$, $\langle \sigma_j \sigma_{j+n} \rangle$ decays exponentially with distance. (e) For $\alpha = 2$, $\langle \sigma_j \sigma_{j+n} \rangle$ decays exponentially with distance.

## 4 EVALUATION

Our experimental goal is to probe the *out-of-the-box*, zero-shot, closed-book performance of general-purpose frontier models without any domain-specific fine-tuning on **CMT-Benchmark**. We evaluate 17 models on the full benchmark, 7 OpenAI's model (GPT-4o, GPT-4.1, GPT-5, GPT-5-mini, GPT-5-nano, GPT-o3, GPT-o4-mini); 3 Google Gemini's models (Gemini 2.0 Flash, Gemini 2.5 Flash, Gemini 2.5 Pro); 5 Anthropic's Claude models (Claude 3.7 Sonnet, Claude 4.0 Sonnet, Claude 4.1 Sonnet, Claude 4.0 Opus, Claude 4.1 Opus); and two open source models (DeepSeek v3, LLaMA

| Model | Overall | HF | ED | DMRG | QMC | VMC | PEPS | SM | Other |
|---|---|---|---|---|---|---|---|---|---|
| GPT-4o | 2.0 | 0.0 | 0.0 | 0.0 | 0.0 | 0.0 | 0.0 | 0.0 | 6.2 |
| GPT-4.1 | 4.0 | 0.0 | 12.5 | 0.0 | 0.0 | 0.0 | 0.0 | 0.0 | 6.2 |
| GPT-5 | 30.0 | 20.0 | 37.5 | 0.0 | 16.7 | 0.0 | 66.7 | 33.3 | 37.5 |
| GPT-5-mini | 24.0 | 20.0 | 37.5 | 0.0 | 16.7 | 0.0 | 33.3 | 50.0 | 18.8 |
| GPT-5-nano | 14.0 | 20.0 | 12.5 | 0.0 | 16.7 | 0.0 | 33.3 | 0.0 | 18.8 |
| GPT-o3 | 26.0 | 20.0 | 50.0 | 25.0 | 16.7 | 0.0 | 66.7 | 16.7 | 18.8 |
| GPT-o4-mini | 18.0 | 20.0 | 25.0 | 0.0 | 16.7 | 0.0 | 33.3 | 33.3 | 12.5 |
| Gemini 2.0 Flash | 10.0 | 20.0 | 25.0 | 0.0 | 0.0 | 0.0 | 0.0 | 16.7 | 6.2 |
| Gemini 2.5 Flash | 4.0 | 20.0 | 12.5 | 0.0 | 0.0 | 0.0 | 0.0 | 0.0 | 0.0 |
| Gemini 2.5 Pro | 14.0 | 20.0 | 12.5 | 0.0 | 0.0 | 0.0 | 33.3 | 0.0 | 25.0 |
| Claude 3.7 Sonnet | 6.0 | 20.0 | 0.0 | 0.0 | 0.0 | 0.0 | 0.0 | 0.0 | 12.5 |
| Claude 4.1 Sonnet | 2.0 | 20.0 | 0.0 | 0.0 | 0.0 | 0.0 | 0.0 | 0.0 | 0.0 |
| Claude 4.0 Sonnet | 6.0 | 20.0 | 12.5 | 0.0 | 0.0 | 0.0 | 0.0 | 16.7 | 0.0 |
| Claude 4.1 Opus | 8.0 | 20.0 | 12.5 | 0.0 | 0.0 | 0.0 | 33.3 | 0.0 | 6.2 |
| Claude 4.0 Opus | 10.0 | 20.0 | 0.0 | 25.0 | 0.0 | 0.0 | 33.3 | 16.7 | 6.2 |
| DeepSeek v3 | 4.0 | 20.0 | 12.5 | 0.0 | 0.0 | 0.0 | 0.0 | 0.0 | 0.0 |
| LLaMA Maverick | 12.0 | 20.0 | 25.0 | 0.0 | 0.0 | 0.0 | 33.3 | 16.7 | 6.2 |

Table 2: Pass@1 rates (%) by model and question type.

Maverick). For the detailed case studies, we refer the reader to Appendix A. Each model is queried with the prompt written by the author, along with a fixed component specifying formatting instructions. In models that support it, we passed system instructions requiring the model to provide a solution in the form of a boxed LaTeX expression. We evaluate the last boxed expression from in response and grade it using the parsing and evaluation code described in Section 3.1.1. The LLM's solution is marked correct if the parser determines it is equivalent to the solution provided by the problem author. Some models, in particular Gemini 2.5 Pro, occasionally disregard the formatting instructions and produce responses that cannot be parsed without a boxed LaTeX environment. The parser failed for a small number of problems, which were human-graded.

We summarize Pass@1 both overall and by topic in Table 2, which reports the results as a percentage correct grouped by model and problem type. Overall accuracy is significantly lower than other physics-related benchmarks (Qiu et al., 2025; Wang et al., 2025). The best performing models were GPT-5 (30.0%), GPT-o3 (26.0%), and GPT-5-mini (24.0%): the only three to score above 20%. The second tier, which we defined as any models with overall $> 10\%$ but below 20%, was comprised of: GPT-o4-mini (18.0%), GPT-5-nano (14.0%), Gemini 2.5 Pro (14.0%), and LLaMA Maverick (12.0%). All other models scored at or below 10%: Gemini 2.0 Flash (10.0%), Claude 4.0 Opus (10.0%), Claude 4.1 Opus (8.0%), Claude 3.7 Sonnet (6.0%), Claude 4.0 Sonnet (6.0%), Gemini 2.5 Flash (4.0%), GPT-4.1 (4.0%), DeepSeek v3 (4.0%), Claude 4.1 Sonnet (2.0%), and GPT-4o (2.0%).

Looking at the results by question type, it is clear that several areas remain extremely challenging for the models to solve. Every model scored 0.0% on VMC problems that required critical judgements while QMC peaks at only 16.7% for the first tier models. Only two models, GPT-o3 and Claude 4.0 Opus, achieved a non-zero score on the DMRG questions (25.0%). In contrast, models did comparatively better with technical questions on using PEPS: the top models reach 66.7% (GPT-5, GPT-o3), and several others achieve 33.3%.

Overall, the results in Fig. 3 show that **CMT-Benchmark** is difficult for even the strongest models. No model approaches mastery across topics. These trends highlight persistent gaps in sign-problem reasoning (QMC), long-range entanglement and boundary effects (DMRG), and symmetry-aware variational projections (VMC).

# 5 CONCLUSION

In this paper, we introduce **CMT-Benchmark**, a benchmark built by expert researchers to mirror real research practice in CMT. The dataset comprises 50 original, expert-authored problems that address methods currently used in quantum and classical CMT research and model-building strategies. The methods we covered are Hartree–Fock, exact diagonalization, density matrix renormalization group, quantum Monte Carlo, variational Monte Carlo, Projected Entangled Pair States, and statistical mechanics. **CMT-Benchmark** expands the current landscape of scientific benchmarks through

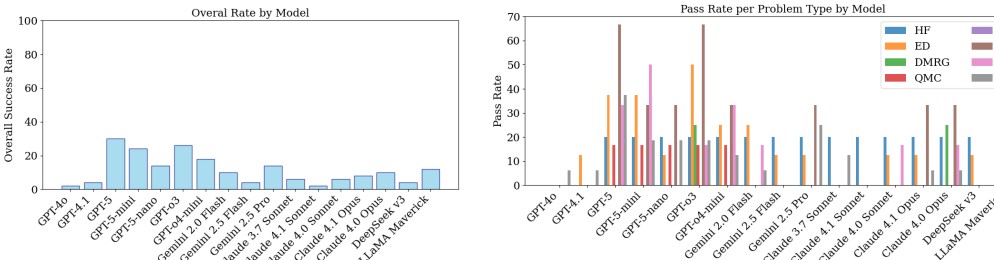

Figure 3: Model performance on **CMT-Benchmark**. (a) Overall success rate on benchmark by model. (b) Success rate per model divided by problem type.

research-grade problems that test LLMs' readiness to work as a research assistant. We tested critical judgment and the ability to synthesize different modalities of information, including mathematical, language-based (conceptual), geometric, and fundamental laws of physics. We evaluated all LLM-generated responses using a deterministic machine-based grading system that supports both numerical and symbolic evaluation, including non-commuting operator algebra. We find that current state-of-the-art LLMs struggle with **CMT-Benchmark**, with the strongest models achieving only 24–30% overall accuracy and no model demonstrating mastery across different problem types.

We gained unique insights through the problem development process that surfaced limitations in frontier LLMs. Since our infrastructure provided a global, real-time view of how models called into the Google Sheet were performing across all the problems, the authors could iteratively identify the angle that caused all the LLMs in the Google Sheet to fail. Firstly, LLMs struggle in connecting 'verbal' expressions to accurate algebraic expressions or geometric ideas. Researchers in the domain can readily translate verbal descriptions, such as "fermionic Hubbard model near half-filling on a Kagome lattice," into an operator algebraic expression for the Hamiltonian. We think in language but calculate using precise mathematical notation. The inability to readily and precisely switch gears between language and mathematics results in LLMs making trivial mistakes and working with expressions that break the laws of physics. This weakness is revealed when problems require answers based on the calculations LLMs must design. Another common struggle is in geometric reasoning. Researchers often sketch geometric view of the problem as a key part of reasoning, as in considering the number of Fermi surfaces in Sec. A.3. The LLM will need to be connected to a tool that can plot the Fermi surface and be instructed on how to do so. Secondly, LLMs struggle with applying fundamental principles such as symmetry to operator algebraic expressions. When relevant terminologies are given, LLMs use the terminologies as a handle to recall the textbook examples. However, a slight departure from the textbook example will trip up the LLM and reveal its limited appreciation for fundamental principles as the foundation of critical judgment. For instance, in a mixed-field Ising model with no $\mathbb{Z}_2$ symmetry to break, some LLMs still predicted a symmetry-breaking transition as a function of the transverse field, misdiagnosing the most basic symmetry structure of the problem. Thirdly, LLMs rely on heuristics when a problem requires a judgment call. For example, in a quantum Monte Carlo efficiency question, LLMs often misattribute the bottleneck in a problem to the so-called 'sign problem'; when the prompt explicitly states the absence of the sign problem, some LLMs can then identify the real bottleneck. Finally, LLMs often fail to recognize the underlying structure or mapping that allows one to leverage known results to simplify the problem. This was revealed in problems that can be mapped to a free fermion problem or problems with an underlying conformal field theory.

## ACKNOWLEDGMENTS

E.-A.K. acknowledges funding support from NSF Award 2433348 and OAC-2118310. H.P. acknowledges the support of US-ONR grant No. N00014-23-1-2357. D. C. is supported in part by a NSF CAREER grant (DMR-2237522), and a Sloan Research Fellowship from the Alfred P. Sloan foundation. T.N. and J.H. acknowledge support from the Swiss National Science Foundation through a Consolidator Grant (iTQC, TMCG-2213805). S.G. acknowledges support from NTT research, the Simons foundation, and a Schmidt sciences polymath award. F.G. acknowledges support from a postdoctoral fellowship of the Stanford's Leinweber Institute of Theoretical Physics. E.K. acknowledges support from the U.S. Department of Energy, Office of Basic Energy Sciences under Grant No.

DE-SC0022311. Y.-J.K. acknowledge support from the NSTC of Taiwan under Grant No. 113-2112-M-002-033-MY3. H.-C.J. is supported by the US Department of Energy, Office of Basic Energy Sciences, Division of Materials Sciences and Engineering, under Contract No. DE-AC02-76SF00515. J.V.R. and M.P.B. acknowledge support from the NSF AI Institute of Dynamic Systems 2112085.

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

## A INSIGHTS

We share some insights from the problem authors on how they designed their problems and the typical failure modes of the LLMs for four different problems below.

### A.1 CASE STUDY: CHARGE DENSITY WAVE IN A TRIANGULAR LATTICE

In the first HF problem from our spreadsheet, we asked for the ground-state energy per site of a commensurate charge density wave on a 2D triangular lattice at half filling with onsite and nearest-neighbor repulsions $U_0$ and $U_1$ in the strong-coupling limit (interaction energy scaling $\gg$ hopping). This problem requires the LLM first to understand the charge ordering in a superlattice embedded in a base triangular lattice. Then it should propose several possible candidate states at half-filling. Finally, it should correctly compute the energy per site and determine the lowest energy state as the ground state. Frontier LLMs consistently failed at the first step to identify the correct ordering in the first place. We show outputs of the four models, Gemini 2.0 Flash, Gemini 2.5 Flash, Gemini 2.5 Pro, and GPT-4o, in the color box below.

> **Model outputs (excerpts) and failure diagnosis**
>
> **Gemini 2.0 Flash:** *In the strong coupling limit, treat hopping as a perturbation and minimize* $E = U_0 \sum_i n_{i\uparrow} n_{i\downarrow} + U_1 \sum_{\langle i,j \rangle} n_i n_j$, *with* $n_i = n_{i\uparrow} + n_{i\downarrow}$. *At half-filling on the triangular lattice, a* $\sqrt{3} \times \sqrt{3}$ *CDW with occupancies* $(0, 1, 2)$ *per tri-sublattice gives* $E_{U_0}/N = U_0/3$ *and (by a balanced-neighbor count)* $E_{U_1}/N = 2U_1$, *hence* $E/N = U_0/3 + 2U_1$. *The uniform state has* $E/N = 3U_1$, *so the CDW is favored.* $\boxed{\frac{U_0}{3} + 2U_1}$
>
> **Gemini 2.5 Flash:** *We need to consider possible ground states that minimize the repulsion.* **Case 1: Uniform charge density (Mott insulator).** *No double occupancy, so* $E_{U_0} = 0$. *With coordination* $z = 6$ *(and thus* $3N$ *bonds),* $E_{U_1} = 3NU_1$, *giving* $E^{(1)}_{per\ site} = 3U_1$. **Case 2: Commensurate CDW** ($\sqrt{3} \times \sqrt{3}$). *Partition into A,B,C sublattices with occupancies* $(2, 1, 0)$ *to maintain average* $n = 1$. *Then* $E_{U_0} = (N/3)U_0$ *and* $E_{U_1} = 2NU_1$, *so* $E^{(2)}_{per\ site} = U_0/3 + 2U_1$. *Comparing, the CDW is favored if* $U_0 < 3U_1$. *... Therefore the commensurate CDW energy per site is* $U_0/3 + 2U_1$. $\boxed{\frac{U_0}{3} + 2U_1}$
>
> **Gemini 2.5 Pro:** *We need to compare the energies of plausible charge configurations. Let* $N_s$ *be the total number of sites.* 1. **Uniform Mott Insulating State:** *One electron per site* ($n_i = 1$ *for all* $i$). 2. **Charge Density Wave (CDW) State:** *The electron density is spatially modulated. At half-filling, a CDW state involves having some sites with zero electrons (empty,* $n_i = 0$) *and some sites with two electrons (doubly occupied,* $n_i = 2$). *To maintain the average filling of one, the number of empty sites must equal the number of doubly-occupied sites* ($N_0 = N_2$). *There can also be singly occupied sites* ($n_i = 1$). *The ground state CDW on a triangular lattice for this model is the* $\sqrt{3} \times \sqrt{3}$ *ordered state. This structure arises from partitioning the lattice into three interpenetrating sublattices (A, B, C), each containing* $N_s/3$ *sites. ... Therefore, this is the ground state energy.* $\boxed{\dfrac{U_0}{3} + 2U_1}$
>
> **GPT-4o:** *Each site, being half-filled, will have an average occupation of one electron. For the onsite repulsion energy, each electron contributes an energy* $\frac{U_0}{2}$, *because the remaining fraction corresponds to preventing double occupancy that provides the remaining* $\frac{U_0}{2}$ *per site.*
>
> All models still fail to display the correct stripe ordering. Gemini 2.0 and GPT-4o proposes a $\sqrt{3} \times \sqrt{3}$; Gemini 2.5 Flash and Gemini 2.5 Pro compare uniform vs CDW and outputs the same expression; The correct ground state is a commensurate stripe CDW at half filling with no double occupancy; in the strong-coupling limit, only $U_1$ contributes, giving $E/N = \boxed{\frac{U_1}{2}}$.

We attribute this to two factors: (i) a language-geometry gap, wherein the model reasons fluently over symbols but does not natively reconstruct 2D lattice structure and commensurability (leading it to suggest incompatible two-sublattice or $\sqrt{3} \times \sqrt{3}$ patterns), and (ii) distributional bias in training data, as triangular-lattice examples are less common than square-lattice ones in standard expositions, and the even the 'stripe phase' embedded in the triangular lattice at half-filling is even less common to appear in the textbook compared to an extended $\sqrt{3} \times \sqrt{3}$ pattern phase, so the model does not robustly internalize the associated counting logic and constraints.

## A.2 CASE STUDY: PHYSICS-INFORMED GROUP THEORY

The second example falls under the ED category. We asked the model to count the number of symmetry-distinct sectors in a Hilbert space, considering a two-dimensional lattice system with 3x3 unit cells, where each unit cell consists of one site that can be empty, occupied by a spin-1/2 electron with either spin, or doubly occupied with two electrons of opposite spin. The system has periodic boundary conditions in both directions. The system has translation symmetry, time-reversal symmetry, and fourfold rotation symmetry. The LLMs' responses are shown below.

---

**Model outputs (excerpts) and failure diagnosis**

**Gemini 2.0 Flash:** *...Time-reversal symmetry means that $H(k) = H(-k)^*$. This doesn't give us any additional symmetry-distinct blocks...* $\boxed{3}$

**Gemini 2.5 Flash:** *...$\mathcal{H}$ contains subspaces with both even and odd $N_f$...* $\boxed{9}$

**Gemini 2.5 Pro:** *We assume that $N_e$ is conserved, so the Hamiltonian is block-diagonal in $N_e$.* $\boxed{86}$

**GPT-4o:** *Counts 9 translation momenta ($\mathbb{Z}_3 \times \mathbb{Z}_3$), includes fourfold rotations ($C_4$), and then heuristically halves by time reversal, concluding there are $\frac{9 \times 4}{2} = 18$ symmetry-distinct blocks.* $\boxed{18}$

In Gemini 2.0 Flash, the LLM completely ignored one of the symmetries (rotation symmetry) and thus could not reason with time-reversal symmetry either. This is a rather blunt failure. In Gemini 2.5 Flash and Pro, both LLMs ignore that particle number or parity conservation was not mentioned as a symmetry in the problem statement, and it was explicitly excluded that non-mentioned symmetries are to be added. Between the two, adding fermion parity conservation is a more subtle mistake, as it is often an underlying assumption in quantum mechanical descriptions without being explicitly stated. The fermion number, however, is frequently not conserved in Hamiltonians of interest, for instance, when describing superconductors. Gemini 2.5 Pro thus fails more drastically. GPT-4o instead multiplies translation and rotation counts and halves by time reversal to obtain 18, which also does not correspond to the correct block structure.

---

The LLMs gravitate towards solving a more standard problem, if a problem appears hard, by changing its assumptions to cases that are more prevalent in the literature (concretely, they have added symmetries). Furthermore, they frequently respond to problem hardness with a lengthy output of dense, nearly cryptic text.

### A.3 CASE STUDY: ENTROPY PRODUCTION AND FLUCTUATION DISSIPATION THEOREM

A third example falls under the Statistical Mechanics category. Here, LLM is asked to consider a peculiar type of overdamped Langevin equation in two dimensions under the action of an external potential, and to determine whether the fluctuation dissipation theorem (Kubo et al., 2012; Groot & Mazur, 2013) is violated. The options available to the model are

- (a) Yes, because the dynamics has a positive entropy production rate.
- (b) No, because the dynamics is time-reversible.
- (c) No, because the Boltzmann distribution is the stationary distribution.
- (d) Yes, because there is a nonzero self-propulsion speed.

The dynamics proposed resemble those of a chiral active Brownian particle (Liebchen & Levis, 2022), which has a net-nonzero entropy production rate and is therefore time-irreversible. However, the frequency of rotation of the self-propulsion direction is chosen in such a way to allow sampling from the Boltzmann distribution with a target temperature and potential energy in the steady state (Ghimenti et al., 2024). Thus, instantaneous snapshots of the particle position look like equilibrium snapshots, even though the dynamics of the particle are out of equilibrium. The simultaneous appearance of these two seemingly contradictory properties is what makes the problem subtle. Indeed, if the steady state of the system has the Boltzmann form, then the fluctuation-dissipation theorem is obeyed. This is a standard result in nonequilibrium thermodynamics (Groot & Mazur, 2013). The correct answer is then (c). However, the models fail to appreciate this connection and focus on the nonreversibility of the dynamics and the presence of a net nonzero entropy production rate. These aspects are ubiquitous in the field of active matter (O'Byrne et al., 2022; te Vrugt et al., 2025), where the fluctuation-dissipation theorem is often violated. Instead of carefully analyzing the proposed dynamics, the models appeal directly to general facts known in the field, leading to an incorrect conclusion.

---

**Model outputs (excerpts) and failure diagnosis**

**Gemini 2.0 Flash:** *The given dynamics is that of an active Brownian particle in two dimensions... Because active Brownian particles consume energy from their environment to maintain their self-propulsion, they are intrinsically out of equilibrium. As such, they have a non-zero entropy production rate in the steady state. This nonequilibrium nature means that the standard fluctuation-dissipation theorem (FDT) is violated.*

**Gemini 2.5 Flash:** *The problem describes an active Brownian particle (ABP) model... the particle consumes energy to maintain its persistent motion. Systems driven by active forces are inherently \*out of thermal equilibrium\* ... the stationary distribution of an active Brownian particle is generally not the Boltzmann distribution...therefore, the system described is unequivocally out of thermal equilibrium. For any nonequilibrium system, the Fluctuation-Dissipation Theorem is generally violated.*

**Gemini 2.5 Pro:** *A system is in thermal equilibrium if it satisfies the principle of detailed balance. In a steady state, this implies that all microscopic probability currents are zero. Active matter systems, by their very nature, are not in thermal equilibrium... Systems with broken detailed balance in a steady state have a continuously positive rate of entropy production... Since the active particle system is in a NESS with a positive entropy production rate, it does not meet the fundamental requirements for the FDT to hold. Therefore, the FDT is violated.*

**GPT-4o:** *Parses the ABP equations and emphasizes that self-propulsion ($v_0 > 0$) drives a non-equilibrium steady state with broken detailed balance and a non-Boltzmann stationary distribution; the antisymmetric **A** term renders the dynamics non-time-reversible. Concludes the FDT is violated.*

All four models directly appeal to general conditions and known properties of active matter systems, without analyzing in depth the specific dynamics proposed. Moreover all the models claim that a positive entropy production rate leads to the violation of the fluctuation-dissipation theorem. Most cases appearing in the literature adhere to this statement, but the implication is not always true, as the proposed problem exemplifies.

---

## A.4 Case study: Fermi surface topology

The model is asked about the number of connected components in the Fermi surface of a given tight-binding model. Specifically, it is a two-dimensional square-lattice tight-binding model with a staggered flux of $\theta$ per plaquette. Although such a tight-binding model has been studied in the literature, the AI models do not seem to be familiar with it. To solve this problem, one needs to write down the Hamiltonian in $k$-space and then diagonalize it to obtain the energy dispersion. All models we tried failed at this step. Although they do realize that we need a unit cell with two sites, resulting in a $2 \times 2$ Hamiltonian in the momentum space, no model has correctly written down the Hamiltonian. It appears that the models fail to visualize the connectivity of the lattice (i.e., which sites are neighbors of each other) and also fail to recognize that the unit vector of the lattice needs to be modified due to the breaking of translation symmetry. When attempting other similar questions, we also observed that the system sometimes cannot correctly identify the topology of the Fermi surface. Solving such problems typically requires humans to visualize the energy dispersion and equal-energy contours, either mentally or by plotting them on a computer. It appears that this kind of spatial reasoning remains a weakness of current models.

---

**Model outputs (excerpts) and failure diagnosis**

**Gemini 2.0 Flash:** *Consider the unit cell as two sites, A and B. The sites are $(n_x, n_y) = (2m, 2n)$ and $(2m + 1, 2n)$. Then $A = (2m, 2n)$, $B = (2m + 1, 2n)$. Then $A \to A + \hat{x}$, $A \to A + \hat{y}$ $B \to B + \hat{x}$, $B \to B + \hat{y}$. The first nearest neighbor to A is A+x. second nearest is A+y which satisfies that $n_x + n_y$ is even. The first nearest neighbor to B is B+x. second nearest is B+y which satisfies that $n_x + n_y$ is even. Let's try this: $t_1 = t_2 = 1$, $\theta = \pi/3$. The dispersion is $E(k) = -2\cos k_x - 2\cos k_y$.*

**Gemini 2.5 Flash:** *This problem exhibits a checkerboard pattern for the y-hopping, which means the unit cell needs to be doubled. We can define a unit cell containing two sites, for example, $(n_x, n_y)$ and $(n_x, n_y + 1)$. However, the y-hopping depends on $n_x$ as well, implying a $2 \times 2$ unit cell in general. A common approach for this type of staggered flux problem is to use a gauge transformation or to double the unit cell in one direction, yielding a $2 \times 2$ Hamiltonian.*
*By applying a suitable gauge transformation (e.g., $c_{x,y} \to e^{i\phi_{x,y}} d_{x,y}$) or by considering a $2 \times 1$ superlattice with primitive vectors $(1, 0)$ and $(0, 2)$, the problem simplifies to a two-band model. The specific gauge transformation $c_{x,y} = d_{x,y} e^{i\frac{\theta}{2}(-1)^x y}$ can simplify the y-hopping. After performing the Fourier transform, the resulting $2 \times 2$ Hamiltonian has the form: $H(\vec{k}) = \begin{pmatrix} -2t_1 \cos k_x & -t_2(1 + e^{i\theta} e^{-2ik_y}) \\ -t_2(1 + e^{-i\theta} e^{2ik_y}) & -2t_1 \cos k_x \end{pmatrix}$.*

**Gemini 2.5 Pro:** *To find $h(\vec{k})$, we sum the hopping terms from a site on sublattice A to its neighboring sites on sublattice B, weighted by phase factors. Let's place an A-site at the origin $(0, 0)$. Its neighbors are at $(1, 0), (-1, 0), (0, 1), (0, -1)$, which are all B-sites. The hopping terms are:*

- *From $(0, 0)$ to $(1, 0)$: $-t_1$. Phase factor: $e^{ik_x}$.*
- *From $(0, 0)$ to $(-1, 0)$: $-t_1$. Phase factor: $e^{-ik_x}$.*
- *From $(0, 0)$ to $(0, 1)$: The starting site $(0, 0)$ has $n_x + n_y = 0$ (even), so the hopping is $-t_2$. Phase factor: $e^{ik_y}$.*
- *From $(0, 0)$ to $(0, -1)$: This is the Hermitian conjugate of the hopping from $(0, -1)$ to $(0, 0)$. The site $(0, -1)$ has $n_x + n_y = -1$ (odd), so the hopping from $(0, -1)$ to $(0, 0)$ is $-t_2 e^{i\theta}$. Thus, the hopping from $(0, 0)$ to $(0, -1)$ is $-t_2 e^{-i\theta}$. Phase factor: $e^{-ik_y}$.*

*Summing these up gives: $h(\vec{k}) = -t_1(e^{ik_x} + e^{-ik_x}) - t_2 e^{ik_y} - t_2 e^{-i\theta} e^{-ik_y} = -2t_1 \cos(k_x) - t_2(e^{ik_y} + e^{-i(k_y + \theta)})$.*

**GPT-4o:** *To solve this problem, we first need to write down the dispersion relation for the given tight-binding model. The tight-binding Hamiltonian on the square lattice with the specified hopping terms can be given as:*

$$H = -\sum_{\vec{r}} \left[ t_1(c_{\vec{r}}^\dagger c_{\vec{r}+\hat{x}} + h.c.) + t_2 \left( \frac{1 + (-1)^{n_x + n_y} e^{i\theta}}{2} \right) (c_{\vec{r}}^\dagger c_{\vec{r}+\hat{y}} + h.c.) \right] - \mu \sum_{\vec{r}} c_{\vec{r}}^\dagger c_{\vec{r}}$$

*The dispersion relation is obtained by performing a Fourier transform:*

$$\epsilon(k_x, k_y) = -2t_1 \cos(k_x) - 2t_2 \left( \frac{1 + (-1)^{n_x + n_y} e^{i\theta}}{2} \right) \cos(k_y) - \mu$$

Gemini 2.0 Flash directly writes down the energy dispersion of the tight-binding model without flux, which suggests the more familiar results misled it. Gemini 2.5 Flash and Gemini 2.5 Pro made the right attempt to construct a $2 \times 2$ Hamiltonian, but failed to identify the correct terms. For example, the hopping from the $A$-site at $(0, 0)$ to the $B$-site at $(1, 0)$ should correspond to an off-diagonal term $\begin{pmatrix} 0 & t_1 e^{ik_x} \\ 0 & 0 \end{pmatrix}$ in the Hamiltonian, which no model produced correctly. GPT-4o was confused about a more basic fact: the coordinates $n_x, n_y$ should not appear after the Fourier transformation to momentum space.

