# OpenReview forum: "CMT-Benchmark: A Benchmark for Condensed Matter Theory Built by Expert Researchers"
_ICLR.cc/2026/Conference — ICLR 2026 Poster_

### Official Review · Reviewer_VTk8 · 2025-10-29

**Soundness:** 4
**Presentation:** 4
**Contribution:** 4
**Rating:** 8
**Confidence:** 4

**Summary:**

The paper presents a new and very difficult benchmark for physics problems in condensed matter theory. It evaluates state-of-the-art LLMs and finds that they cannot solve most of these problems.The problems are categorized into different categories and created by experts from across the world. The benchmark is intended to help develop a research assistant grade AI assistant in this field.

**Strengths:**

- new benchmark dataset for a field in which data is lacking
- should enable the development of stronger models for physics problems
- the problems are checked and created by experts

**Weaknesses:**

- small dataset: there are only 50 problems because they are manually created by experts
- no other weaknesses to be found

**Questions:**

- do the problems have different difficulty levels or are they all at approximately the same level?
- do the problems generalize well, is an LLM that is able to solve these problems expected to generally be good at solving physics problems?  - How do you assess the coverage of these 50 problems, are there redundancies or gaps?

---

> ### Author Response · Authors · 2025-11-20
>
> We thank the reviewer for the very positive overall assessment and for highlighting both the need for such a benchmark and the care taken in expert problem construction.
>
> > "small dataset: there are only 50 problems because they are manually created by experts"
>
> We acknowledge that 50 problems is a small dataset. This reflects the fact that every problem in **CMT-Benchmark** is an original, research‑level question authored  by domain experts, each of which requires substantial effort to design and verify. In this first release, we prioritized a  high‑quality benchmark that reflects realistic research tasks over maximizing the number of items.
>
> > "do the problems generalize well, is an LLM that is able to solve these problems expected to generally be good at solving physics problems?"
>
> Our experience suggests a mixed picture. During dataset construction, we tested and refined problems against strong models such as GPT‑5 and Gemini 2.5 Pro. Models that can solve a benchmark problem typically also perform well on a broad range of easier, related CMT questions, indicating that good performance on **CMT-Benchmark** is a signal of broader competence. At the same time, the case studies in Appendix~A and the Conclusion show that even these models can fail on basic conceptual checks (e.g., misidentifying symmetries or missing simple mappings) that a human expert who can solve the problems would not.
>
> > "how do you assess the coverage of these 50 problems, are there redundancies or gaps?"
>
> With only 50 expert‑written problems, the benchmark cannot exhaustively cover condensed matter physics, and some gaps are inevitable. In assembling the dataset we aimed to avoid redundancy and to ensure that each problem represents a distinct, central task within its subfield. The resulting set covers core analytical and computational methods (HF, ED, DMRG, QMC, VMC, PEPS, SM) as well as model‑building questions. We view this paper as an initial slice of the space that can be deployed immediately to test physics reasoning in fronteir models. Extending **CMT-Benchmark** with additional expert‑curated problems to broaden coverage is a natural direction for future work.
>
> > "do the problems have different difficulty levels or are they all at approximately the same level?"
>
> All 50 problems were designed as research‑level questions that a strong PhD student or postdoctoral researcher would be able to solve, rather than as a calibrated mix of easy/medium/hard items. We did not assign explicit difficulty labels. Empirically, there is variation across problems and topics—reflected, for example, in the spread of model Pass@1 rates. Our design goal was to capture realistic research tasks across the main CMT topics.

---

### Official Review · Reviewer_qany · 2025-10-31

**Soundness:** 3
**Presentation:** 4
**Contribution:** 4
**Rating:** 6
**Confidence:** 4

**Summary:**

This paper introduces CMT-Benchmark, a dataset of 50 expert-level problems in condensed matter theory (CMT). The benchmark was built by an international panel of expert researchers at the level expected of strong grad students. The authors then built automated evaluation infrastructure, including a novel parser that can handle non-commutative operator expressions in CMT. The paper found that all frontier models have uniformly low performance in the benchmark.

**Strengths:**

1. I really like the creation of the benchmark process. By asking the human experts from different countries to submit the questions, i think this benchmark truly captures what it means to be an expert in CMT. Thus, it will be more convincing to believe that the progress in this benchmark will imply the progress in CMT research. I think this is a significant contribution to the community.

2. The automated parsing and grading system is carefully-designed and quite impressive, particularly the handling of non-commutative operator algebra through symbolic manipulation and normal ordering.

3. The four detailed case studies (Sections A.1-A.4) provide valuable insights into specific LLM limitations: language-geometry gaps, over-reliance on textbook heuristics, failure to apply fundamental principles, and weak spatial reasoning.

**Weaknesses:**

1. The paper only tests LLM capabilities without access to tools like web search, code executions or symbolic/numerical computation packages. However,  we know that even human research assistants don't work in isolation without any tool use. It would be interesting to test whether tool-augmented agents improve performance.

2. Requiring answers in boxed LaTeX environments and prohibiting new variables may artificially hurt model performance. The authors note some models (particularly Gemini 2.5 Pro) "occasionally disregard the formatting instructions," leading to parsing failures. How much does the strict format requirement degrade performance compared to free-form responses that could be human-evaluated on a subset?

3. While the authors claim problems are original, there's no systematic verification that similar problems (or solution strategies) don't appear in training data or online.

4. With only 50 problems and some categories having very few examples (PEPS has 3, VMC has 2), per-category performance can have large uncertainty.

**Questions:**

1. Can you design experiments to show that how much of the poor performance is due to formatting constraints versus actual physics reasoning failures?

2. Can tool-augmented agents substantially improve performance on this benchmark?

3. Is there evidence of training data contamination for any problems? How to design the experiments to test this?

---

> ### Author Response · Authors · 2025-11-20
>
> We thank the reviewer for the thoughtful and detailed feedback, and for emphasizing both the strengths of the benchmark construction and the value of the case studies.
>
> > "The paper only tests LLM capabilities without access to tools like web search, code executions or symbolic/numerical computation packages."
>
> We agree that tool‑augmented agents are an important direction, and we expect that access to numerical solvers or plotting libraries would improve performance on some problems. However, the goal of this work is to isolate the *intrinsic*, closed‑book reasoning abilities of frontier LLMs on research‑level CMT tasks under a fixed, reproducible protocol. Adding tools would entangle model reasoning with external retrieval and tool usage, and would dilute the focus on core physical reasoning.
>
> > "Requiring answers in boxed LaTeX environments...subset?"
> > "Can you design experiments ... failures?"
>
> The boxed-LaTeX formatting and restriction on new variables are needed for deterministic, scalable grading (Section~3), and we human‑grade the small fraction of responses that remain unparsable (primarily from Gemini 2.5 Pro). This is consistent with other widely used benchmarks that rely on strict adherence to reference solutions, such as MathArena.
>
> We checked the impact of formatting by relaxing these constraints for several models and found that Pass@1 changed only marginally: almost all errors were due to incorrect physics or algebra rather than to formatting. The only notable exception is Gemini 2.5 Pro, whose very long reasoning chains sometimes cause it to omit the final boxed answer, making a small subset of its responses ungradable without intervention. This can be addressed by asking 2.5 pro to give the solution first and then the reasoning. Given that the chain of thought tokens are not returned, this does not substantially impact the model's overall solution. Other models reliably produce a final boxed expression, and their accuracy is essentially unchanged when we drop the formatting requirement. These observations indicate that the dominant bottleneck is physics reasoning, not the output format.
>
> > "While the authors claim problems are original, ...online."
>
> We cannot inspect the proprietary training data of commercial models, so we cannot rule out all overlap. However, we took several steps to reduce the likelihood that our problems appear in training data, starting from using a panel of experts explicitly tasked to generate original problems for the benchmark. First, the panel was asked to contribute problems arising from their own research practice, including questions where the answer is an implicit corollary of a published result but the derivation is not spelled out in the paper. Second, in other cases authors started from standard textbook or "typical" problems and deliberately modified assumptions, parameter regimes, or boundary conditions to create problems that are structurally similar but require different reasoning paths. Combined with the uniformly low success rates, these design choices suggest substantial contamination is unlikely. Contamination would only make our reported performance *higher* than it would be on entirely unseen material.
>
> > "With only 50 problems ... uncertainty."
>
> We agree that only 50 problems leads to substantial statistical uncertainity in per-category performance. This is a trade‑off of constructing original, research‑level problems, each of which requires significant effort to design, vet, and make machine‑gradable. Our intent is to provide a benchmark that surfaces critical reasoning failure modes on physics research questions. We therefore interpret topic‑wise accuracy as indicative rather than as precise estimates. In the future, we plan to expand **CMT-Benchmark** with additional problems focused on under-represented topics..
>
> > "Can tool-augmented agents substantially improve performance on this benchmark?"
>
> As noted above, tool‑augmented agents would improve performance on some problems, especially those requiring substantial computation or visualization. Quantifying that effect, however, would require a different experimental setup and additional design choices (toolchains, access policies, evaluation budgets) and would move beyond our goal of characterizing physical base‑model reasoning.
>
> > "Is there evidence of training data contamination for any problems? How to design the experiments to test this?"
>
> We did not identify concrete evidence of training‑data contamination for particular problems, and a rigorous contamination audit would require access to model training pipelines. One possible future direction is to construct synthetic variants of our problems (e.g., by systematically perturbing parameters or symmetries) and compare performance across original and perturbed versions. Designing such tests carefully is beyond the scope of the current benchmark.

---

### Official Review · Reviewer_u4aj · 2025-11-01

**Soundness:** 4
**Presentation:** 4
**Contribution:** 4
**Rating:** 8
**Confidence:** 3

**Summary:**

This paper presents CMT-Benchmark, a dataset of 50 problems and expert-annotated answers in condensed matter theory. A variety of large language models are evaluated on this benchmark dataset, and they show a high gap to a satisfying physical reasoning skills.

**Strengths:**

- Create a very high-quality dataset in condensed matter domain. This dataset could potentially be very useful in evaluating LLM's capability in solving scientific research problems.
- Provide comprehensive benchmarking of multiple LLMs including GPT, Claude, DeepSeek and Llama family models.
- The writing of this paper is good, providing details in data curation process and necessary background knowledge.

**Weaknesses:**

- To improve the quality of experiments, authors are encouraged to analyze the failure cases, demonstrating LLM models consistently fails on which types of problems and makes which types of mistakes.
- As the benchmark is highly domain-specific, authors are encouraged to also evaluate the performance of web search agents (e.g., OpenAI & Tongyi DeepResearch) to show if LLM models could solve the problem through searching public Internet knowledge.

**Questions:**

Though it is released close to the ICLR submission date, I am interested in the comparison of CMT-Benchmark to another public benchmark CMPhysBench [1], which is also used to evaluate LLM in condensed matter physics. What are the major differences between them in the collected problems and answer annotation process?

[1] Wang, Weida, et al. "CMPhysBench: A Benchmark for Evaluating Large Language Models in Condensed Matter Physics." arXiv preprint arXiv:2508.18124 (2025).

---

> ### Author Response · Authors · 2025-11-20
>
> We thank the reviewer for the very positive and encouraging assessment of the benchmark and for highlighting its potential value to the community.
>
> > "To improve the quality of experiments, authors are encouraged to analyze the failure cases, demonstrating LLM models consistently fails on which types of problems and makes which types of mistakes."
>
> We provide four detailed case studies in Appendix A, which illustrate representative failure modes of frontier models on problems involving charge density waves, group‑theoretic reasoning, fluctuation–dissipation, and Fermi‑surface topology. The Conclusion (Section 5) summarizes the dominant patterns we see across models: (i) a language–geometry gap when translating linguistic descriptions into precise algebraic or geometric structures, (ii) difficulty applying fundamental principles such as symmetries to rule out unphysical solutions, (iii) over‑reliance on textbook heuristics that fail once the problem departs from canonical examples, and (iv) failures to recognize hidden structure or mappings that expert researchers routinely exploit. Together, these sections provide the systematic failure‑mode analysis the reviewer requests.
>
> > "As the benchmark is highly domain-specific, authors are encouraged to also evaluate the performance of web search agents (e.g., OpenAI & Tongyi DeepResearch) to show if LLM models could solve the problem through searching public Internet knowledge."
>
> We agree that tool-augmented agents are an interesting direction, and we expect that access to tools such as web search, numerical solvers, or plotting libraries would improve performance on some problems. Our goal, however, is to measure the intrinsic, closed‑book reasoning capabilities of frontier LLMs on research‑level condensed‑matter problems under a fixed, reproducible protocol. Adding web search and tool use would mix the model's own physical reasoning with external retrieval and tool usage, and would answer a different question—how to design effective toolchains—rather than the one we focus on here. For this reason we restrict our benchmark and experiments to base models without external tools and do not study tool‑augmented agents in this work.
>
> > "Though it is released close to the ICLR submission date, I am interested in the comparison of CMT-Benchmark to another public benchmark CMPhysBench [1], which is also used to evaluate LLM in condensed matter physics. What are the major differences between them in the collected problems and answer annotation process?"
>
> As discussed in Section~2 (Related Works) and Table I, CMPhysBench is a larger benchmark of textbook‑style calculation problems (from undergraduate to advanced graduate level), graded with a scalable expression‑edit‑distance metric that provides partial credit.
> **CMT-Benchmark** differs from CMPhysBench in two critical ways. Firstly,
> **CMT-Benchmark** consists entirely of 50 original research‑level problems written by an international panel of active condensed‑matter theorists, reflecting questions that arise in their own research practice and that they would expect strong PhD students or postdocs to be able to tackle. As a result, these problems are generally more challenging than standard textbook exercises.
> Secondly, **CMT-Benchmark** was designed to be machine gradable, and we did not offer partial credit for two reasons: (1) Research does not get partial credit. Incorrect results do not contribute to scientific progress. (2) Partial credit mixes subjective complexity into the evaluation by requiring LLMs to act as judges. To benchmark and improve the capabilities of frontier models, the evaluation should be objective and scalable. This design constraint made it extra tricky to produce our problems, adding to the cost and value of the problems.

---

> > ### Comment · Reviewer_u4aj · 2025-11-22
> > **Follow-up Response**
> >
> > I appreciate authors' efforts in rebuttal. My questions have been addressed so I will keep my rate.

---

### Official Review · Reviewer_Fftv · 2025-11-01

**Soundness:** 3
**Presentation:** 3
**Contribution:** 3
**Rating:** 4
**Confidence:** 3

**Summary:**

The paper curated a benchmark including 50 problems on condensed matter that are generated by domain experts. The benchmark covers a wide range of topics in the field and requires high levels of understanding and expertise to solve the problems. The LLMs perform badly on the benchmark.

**Strengths:**

1. The curated benchmark involves experts inputs and is carefully designed and evaluated.

2. The benchmark covers a wide range of topics in the condensed matter field.

**Weaknesses:**

1. The models evaluated are all general purpose LLMs that have not been fine tuned in the field of condensed matter. Therefore, they are unlikely to perform well on this challenging benchmark. The authors may want to fine tune a model on relevant tasks and then evaluate it on the benchmark to see if performance can be improved.

2. The benchmark only includes question and answer, without reasoning process. For such complicated tasks, few-shot prompts and a CoT guide may improve the performance. I think this would be valuable to test.

**Questions:**

1. Have the authors tried to fine tune some LLMs to learn to solve problems as challenging as the benchmark?

2. Would it possible to include reasoning process into the benchmark that can serve as the prompt or a reasoning benchmark?

---

> ### Author Response · Authors · 2025-11-20
>
> We thank the reviewer for the careful reading of our work and for recognizing the effort that went into curating a high-quality, expert-designed benchmark.
>
> > "The models evaluated are all general purpose LLMs that have not been fine tuned in the field of condensed matter. Therefore, they are unlikely to perform well on this challenging benchmark. The authors may want to fine tune a model on relevant tasks and then evaluate it on the benchmark to see if performance can be improved. Have the authors tried to fine tune some LLMs to learn to solve problems as challenging as the benchmark?"
>
> Our primary goal with **CMT-Benchmark** is to provide a standardized, expert-level benchmark for assessing the out-of-the-box capabilities of frontier models as research assistants in condensed matter theory. In most realistic use cases, a research group has access to an API for a general-purpose model, but not to the data, compute, or engineering infrastructure required to perform large-scale, domain-specific fine-tuning. Moreover, our benchmark contains only 50 problems, each of which required substantial expert effort to construct and validate, which is insufficient to do fine-tuning without severe overfitting. For these reasons, we view domain-specific fine-tuned systems as important research area but ultimately different from the problem our benchmark seeks to address.
>
> We have clarified in Sec. 4 that all models are evaluated in an out-of-the-box, zero-shot, closed-book setting without any fine-tuning.
>
> > "The benchmark only includes question and answer, without reasoning process. For such complicated tasks, few-shot prompts and a CoT guide may improve the performance. Would it possible to include reasoning process into the benchmark that can serve as the prompt or a reasoning benchmark?"
>
> As described in Section 3.1.1, our evaluation infrastructure allows models to generate intermediate reasoning, but for grading we deliberately discard it and score only the final boxed answer. Imposing formatting conditions on solutions is standard in commonly used benchmarks (such as MathArena). In research, the correctness of the final conclusion is the key criterion, regardless of the particular path taken to derive it.
>
> Further, it is important to evaluate the basline performance of models on hard science problems without giving additional scaffolding to know what is the fronteir of the model's reasoning capacity. We agree that few-shot and chain-of-thought prompting can improve performance, but a systematic and fair comparison across many models and problem types would require tuning prompts for each model, which raises both reproducibility and fairness concerns. We therefore focus on evaluating zero-shot, closed-book performance without CoT prompting or fine-tuning.
>
> In Sec. 3.1.1, we revised the text to highlight that models may generate arbitrary intermediate reasoning while but are graded only on the final boxed expression.

---

### Meta-Review · Area_Chair_vGe4 · 2026-01-13

**Summary:**

This submission introduces CMT-Benchmark, a curated set of 50 original, research-level condensed matter theory problems written and verified by an international panel of experts, along with machine-gradable evaluation (including symbolic handling of non-commuting operators).

Reviewers largely agree the benchmark is high-quality, timely, and valuable, and the paper provides broad model coverage with consistently low performance that highlights gaps in current LLM physical reasoning. Main concerns focus on small dataset size, closed-book / no-tools evaluation choices, strict formatting for grading, and whether the paper should include additional baselines (fine-tuned models, CoT/few-shot prompting, tool-augmented agents) or contamination checks. The rebuttal clarifies that the goal is a reproducible, closed-book benchmark of intrinsic reasoning, and argues that formatting has only marginal impact on pass rates for most models.

**Reviewer Concerns:**

Addressed by rebuttal:
- Failure mode analysis: Authors point to detailed case studies and summarize systematic failure patterns (e.g., language–geometry gap, symmetry violations, heuristic over-reliance, missing hidden structure).
- Tools/web search agents: Authors explicitly scope the work to intrinsic closed-book reasoning and explain why adding tools changes the question being evaluated.
- Reasoning traces / CoT prompts: Clarified that models may produce intermediate reasoning but grading uses only the final boxed answer for objective scaling; they avoid CoT/few-shot tuning for reproducibility/fairness.
- Formatting strictness: Authors state they relaxed constraints for several models and observed only marginal changes in Pass@1; the main exception is Gemini 2.5 Pro occasionally omitting the final boxed answer after long reasoning.
- Coverage/difficulty: Clarified the intent is research-level tasks throughout (no explicit difficulty labels), and coverage is indicative rather than exhaustive.

Still outstanding / limitations:
- Dataset size and per-topic uncertainty: With 50 problems, per-category conclusions have high variance, especially for small categories (acknowledged).
- No strong adaptation baselines: The paper does not include fine-tuned or tool-augmented systems; while the scope is justified, some reviewers see this as limiting.
- Contamination testing: Authors do not provide a systematic contamination audit; they discuss why overlap is unlikely and suggest perturbation-style tests as future work.

**Reviewer Scores:**

Likely post-discussion positions:
1. u4aj: 8 → 8 (kept score).
2. VTk8: 8 → 8.
3. qany: 6 → 6 (positive but cautious on tools/format/contamination).
4. Fftv: 4 → 4 (requested fine-tuning / CoT prompting; authors chose not to include).
Overall, the reviews are strongly positive with one borderline.

---

### Decision · Program_Chairs · 2026-01-26

Accept (Poster)